# Analysis of the Spatial and Temporal Distribution of Process Gases within Municipal Biowaste Compost

**Sylwia Stegenta** [1] , **Karolina Sobieraj** [1] , **Grzegorz Pilarski** [2] , **Jacek A. Koziel** [3] and **Andrzej Białowiec** [1,]*

1   Institute of Agricultural Engineering, Faculty of Life Sciences and Technology, Wroclaw University of Environmental and Life Sciences, 51-603 Wroclaw, Poland; sylwia.stegenta@upwr.edu.pl (S.S.); karolina.sobieraj@upwr.edu.pl (K.S.)

2   Best-Eko Sp. z o.o. Company, 44-240 Żory, Poland; grzegorz.pilarski@best-eko.pl

3   Department of Agricultural and Biosystems Engineering, Iowa State University, Ames, IA 50011-3270, USA; koziel@iastate.edu

*   Correspondence: andrzej.bialowiec@upwr.edu.pl; Tel.: +48-713205700

**Abstract:** Composting processes reduce the weight and volume of biowaste and produce products that can be used in agriculture (e.g., as fertilizer). Despite the benefits of composting, there are also problems such as odors and the emission of pollutants into the atmosphere. This research aimed to investigate the phenomenon of process gas ($CO$, $CO_2$, $NO$, $O_2$) evolution within a large-scale municipal composter. The effects of turning frequency and pile location (outdoor vs. indoors) on process gas and temperature spatial and temporal evolution were studied in six piles (37-81 tons of initial weight) over a six-month period. The biowaste consisted of green waste and municipal sewage sludge. The chemical composition and temperature of process gases within four cross sections with seven sampling locations were analyzed weekly for ~7–8 weeks (a total of 1375 cross sections). The aeration degree, temperature, $CO$, $CO_2$, and $NO$ concentration and their spatial and temporal distribution were analyzed. Final weight varied from 66% reduction to 7% weight gain. Only 8.2% of locations developed the desired chimney effect (utilizing natural buoyancy to facilitate passive aeration). Only 31.1% of locations reached thermophilic conditions (necessary to inactivate pathogens). Lower $O_2$ levels corresponded with elevated $CO_2$ concentrations. $CO$ production increased in the initial composting phase. Winter piles were characterized by the lowest $CO$ content. The most varied was the $NO$ distribution in all conditions. The $O_2$ concentration was lowest in the central part of the pile, and aeration conditions were good regardless of the technological regime used. Turning once a week was sufficient overall. Based on the results, the most favorable recommended procedure is turning twice a week for the first two weeks, followed by weekly turning for the next two weeks. After that, turning can be stopped unless additional removal of moisture is needed. In this case, weekly turning should continue until the process is completed. The size of the pile should follow the surface-to-volume ratio: <2.5 and <2 for cooler ambient conditions.

**Keywords:** biowaste; sewage sludge; waste management; composting; aeration; carbon monoxide; greenhouse gases; temperature; turned piles; spatial distribution

---

## 1. Introduction

Composting processes reduce the weight and volume of waste and produce a useful line of products that can be used in agriculture (e.g., as fertilizer). Despite the benefits of this process, there are also problems such as odors and the emission of pollutants into the atmosphere [1]. Composting uses carbon-rich waste that generates air pollutants such as $CO$, $CO_2$, $NH_3$, $NO$, $NO_2$, $SO_2$, and $H_2S$ [2].

Gases generated in compost can be acutely and chronically toxic (e.g., CO and $H_2S$), posing a risk to plant operators. Emissions of other gases produced during the biological degradation of organic matter such as $CH_4$ and $H_2$, as well as nitrogenous compounds and volatile organic compounds (VOCs), are also documented [3]. Process gas generation is confounded by many interdependent variables that have unique spatial and temporal patterns. Thus, it is challenging to sustainably manage large composting operations with variable biowaste inputs and environmental constraints.

$O_2$ is essential for microbial activity in composting due to the aerobic nature of the process [4]. On the one hand, an insufficient aeration rate causes undesirable anaerobic conditions [5]. The minimum recommended $O_2$ concentration in the pores of the compost material is ~5% [6]. However, too much aeration results in excessive heat and vapor loss, decreased temperature, condensation of leachate, and stress to microbial activity and the entire degradation process. Excessive aeration ($O_2$ level > 15%) causes absorption of heat from the outer layers of the material. The amount of available $O_2$ in the compost depends on many factors, e.g., the type of material, porosity, particle size, depth and location within the pile, temperature, and humidity level [7]. Operational parameters such as the aeration rate, turning frequency, and turning schedule can also affect $O_2$ levels. Haga et al. [8] determined the $O_2$ zone within 20-30 cm from the surface of the pile. This $O_2$ stratification may also influence the spatial and temporal distribution of other process gases (e.g., CO, $CO_2$, $CH_4$, NO) and temperature within the pile.

CO is still rarely reported in studies of the composting process. It is a primary ambient air pollutant that is not typically associated with sustainable practices such as composting biowaste. CO can react with hydroxyl radical and increasing $O_3$ and $CH_4$ levels in the troposphere [9]. The biological production of CO is still little known [3]. Stegenta [10] showed that the biological production of CO in full-scale composting could occur in mesophilic compost conditions up to 40 °C. At higher temperatures, CO is oxidized biologically by thermophilic microorganisms. Maximum concentrations of ~160 ppm (in the case of organic waste), ~120 ppm (in the case of garden waste), and 10 ppm (in the case of animal waste) were reported for lab- and full-scale composting, respectively [11]. Stegenta et al. [2] reported CO emission rates from the composting of municipal solid waste (MSW) as high as 143 $mg \cdot t^{-1} \cdot h^{-1}$. The presence of CO showed a positive correlation with the availability of $O_2$, which confirms the importance of aeration conditions. Also, CO accumulation in piles usually occurs when $O_2$ is present [9,11]. The CO release rate is higher in the initial phase of composting, which is associated with the rise in temperature. CO emissions decrease with the compost maturation, which is caused by the consumption of CO by microorganisms [12].

$CO_2$ emissions can also vary during composting. $CO_2$ production is high in the initial phase of the process when easily degradable substrates are utilized and the microbial activity is high. The highest $CO_2$ emissions occur at elevated temperatures, associated with thermophilic bacteria activity [13]. The intensity of $CO_2$ emissions decreases when the temperature drops below ~40 °C and the composting process enters the maturation phase [14]. $CO_2$ emissions from compost can be considered non-fossil carbon emissions. The justification is based on the conviction that the climate 'responds' equally irrespective of the source and should be accounted for similarly [15]. Assuming this is the case, composting processes could have a relatively high impact on air pollution.

The NO emissions potential from compost piles is not fully understood [16]. Studies on NO emissions from composting processes are also rare. NO emissions are mainly considered in the context of the N cycle for soils [17]. NO emissions can be kept low by excessive aeration yet they cannot be avoided [18]. NO accumulates in anaerobic zones, which is common in the early phase of composting and aeration. The NO production is eventually reduced when longer aeration is carried out. Studies have not shown an effect of pH on NO emissions [19]. On the other hand, the apparently opposite effect on NO and CO production that is controlled by $O_2$ levels emphasizes the need to carefully consider process gases comprehensively so that some optimization could mitigate emissions. Also, the apparent importance of the early phase of composting on the generation of CO, NO, and $CO_2$ suggests that some phase-based optimization might be needed.

Compost temperature is also linked to the aeration regime. The literature describes a desired 'chimney effect,' i.e., when the warm air from the compost core naturally rises and leaves the top of the pile due to buoyancy while drawing in cooler (ambient) air from the sides near ground [20]. This air flow is in turn related to the amount of free (pore) space inside the compost [21]. The chimney effect occurs in systems with both natural and forced aeration. For instance, when composting in a bioreactor with bottom aeration, temperature gradients can be seen depending both on depth and distance from side walls. The temperatures near the core and in the upper part of the compost pile are the highest; the coldest is near the bottom and at the sides [22]. This may be due to dead spaces, from a few to several centimeters thick, which cannot be reached by the mixer blades during turning. These spaces act as an insulator and inhibit heat transfer [23].

Monitoring the compost process completion can also be achieved via analysis of the temporal and spatial distribution. Researchers have reported the use of specific markers (typically volatile organic compounds, VOCs) that are associated the progression of compost biomass decay or with the digestion of animal carcasses inside compost material generally available on farms [24–28] or decaying due to forced aeration [29,30].

Sustainable composting technology requires an improved understanding of both positive and negative aspects such as the generation air pollution. The increased availability of quantitative data on process gas emissions from composting plants, including greenhouse gases (GHGs), can improve the comparability, coherence, and accuracy of emission inventories reported in national and international databases [31]. Therefore, there is a need for research in this area, especially when the selective collection of biowaste is developing or mandatory in many EU countries, including Poland [32].

Poland generates >0.5 million $Mg \cdot y^{-1}$ (dry basis) of sewage sludge, of which ~56% is stored (in lagoons at wastewater treatment plants) for further utilization, which involves agricultural usage (38%), thermal treatment (13.5%), and landfilling (6%) [33]. Due to the high moisture content in sewage sludge, the thermal treatment may be characterized by high energy demand, which is why this method of management is dedicated mainly to large wastewater treatment plants. For small- and medium-sized facilities, composting prior to agricultural use is recommended. In the case of agricultural applications, the requirements for heavy metal content in sludge management should be met [34].

Research on the accumulation of process gases during composting in turned (managed) piles is particularly important as municipalities are required to utilize the organic fraction of MSW. These requirements are consistent with the 'zero waste' and phased-out landfilling in EU policy goals [35–37]. Large composting operations are in place utilizing other municipal waste streams, e.g., sewage sludge and yard waste (grass clippings, leaves, branches). Knowledge about the spatial and temporal distribution of process gases can enable optimization and adoption of the biowaste turning regime and control of parameters such as temperature or $O_2$. Aeration can be optimized and be considered for improved management process gases, their emissions, and the overall composting impact on the environment.

The research aimed to investigate the phenomenon of process gas (CO, $CO_2$, NO, $O_2$) spatial and temporal evolution during large-scale composting of biowaste depending on the piles' turning regime (no turning, turning once a week, or turning twice a week), and location (outdoors or inside a composting hall). The process gas chemical composition and determination of temporal and spatial variability of concentrations and temperatures were performed at a large, municipal composting plant.

## 2. Materials and Methods

### 2.1. Characteristics of Composted Biowaste

Biowaste compost piles were a mixture of grass, leaves, and branches delivered to the composting plant by residents of Rybnik, as well as dehydrated and stabilized sewage sludge coming from the "Boguszowice" wastewater treatment plant (3270 $m^3 \cdot day^{-1}$ municipal sewage), Silesian Voivodship, Poland. Piles were characterized by an equal volume ratio of components (4 parts green waste

(GW)—grass and leaves, 2 parts branches (BR), and 1 part sewage sludge (SS)). The composting process was carried out on a large plant scale—the initial weight of the pile ranged from 36,960 to 70,880 kg (Table 1). The mass composition varied due to the different bulk weight of the waste used for the process, as shown in Table 1. The applicable international standards for moisture, loss on ignition (LOI), and bulk density (PN-EN 14346:2011; PN-EN 15169:2011; PN-EN ISO 17828:2016-02, respectively [38–40]) were adhered to (Table 2). Waste stability indicator-respiration activity for four days ($AT_4$) was determined in OxiTOP (WTW, Weilheim in Oberbayern, Germany) equipment according to Binner et al. [41].

**Table 1.** Mass composition of the analyzed piles before and after the composting process.

| Pile | Biowaste Type | Ingredients Mass | Percent of Total Mass | Pile Mass before the Process | Pile Mass after the Process | Mass Reduction |
|------|---------------|------------------|-----------------------|------------------------------|-----------------------------|----------------|
|      |               | kg               | %                     | kg                           | kg                          | %              |
| A1   | SS            | 20,220           | 28.5                  | 70,880                       | 24,320                      | 65.7           |
|      | GW            | 28,720           | 40.5                  |                              |                             |                |
|      | BR            | 21,940           | 31.0                  |                              |                             |                |
| A2   | SS            | 19,600           | 23.9                  | 81,860                       | No data                     | No data        |
|      | GW            | 48,540           | 59.3                  |                              |                             |                |
|      | BR            | 13,720           | 16.8                  |                              |                             |                |
| A3   | SS            | 19,400           | 39.1                  | 49,580                       | 35,420                      | 28.6           |
|      | GW            | 19,740           | 39.8                  |                              |                             |                |
|      | BR            | 10,440           | 21.1                  |                              |                             |                |
| A4   | SS            | 6680             | 15.9                  | 42,030                       | 39,960                      | 4.9            |
|      | GW            | 22,550           | 53.7                  |                              |                             |                |
|      | BR            | 12,800           | 30.5                  |                              |                             |                |
| A5   | SS            | 6620             | 17.6                  | 37,520                       | 34,100                      | 9.1            |
|      | GW            | 21,220           | 56.6                  |                              |                             |                |
|      | BR            | 9680             | 25.8                  |                              |                             |                |
| A6   | SS            | 6120             | 16.6                  | 36,960                       | 39,710                      | −7.4           |
|      | GW            | 20,920           | 56.6                  |                              |                             |                |
|      | BR            | 9920             | 26.8                  |                              |                             |                |

Note: SS = sewage sludge, GW = green waste (grass and leaves), BR—tree branches.

**Table 2.** Physicochemical properties of compost pile materials.

| Pile | Biowaste Type | Moisture | | Loss on Ignition | | | Bulk Density | |
|------|---------------|----------|------------------------------------------|------------------|---------|------------------------------------------|--------------|------------------------------------------|
|      |               | %        | Weighted Average (±Standard Deviation)   | % DM             | Average | Weighted Average (±Standard Deviation)   | kg·m$^{-3}$  | Weighted Average (±Standard Deviation)   |
| A1   | SS            | 73.83    |                                          | 49.13 49.62      | 49.57   |                                          | 624          |                                          |
|      | GW            | 51.89    | 53.0 ± 19.33                             | 47.21 49.58 46.41 | 47.73   | 55.3 ± 12.64                             | 194          | 339 ± 230.5                              |
|      | BR            | 35.3     |                                          | 69.30 70.32      | 70.49   |                                          | 265          |                                          |
| A2   |               |          |                                          | No data          |         |                                          |              |                                          |

**Table 2.** *Cont.*

| Pile | Biowaste Type | Moisture | | Loss on Ignition | | | Bulk Density | |
| | | % | Weighted Average (±Standard Deviation) | % DM | Average | Weighted Average (±Standard Deviation) | kg·m$^{-3}$ | Weighted Average (±Standard Deviation) |
|---|---|---|---|---|---|---|---|---|
| A3 | SS | 63.85 | | 31.83 31.96 | 31.32 | | 859 | |
| | GW | 36.18 | 46.9 ± 16.05 | 57.65 59.98 | 58.90 | 47.6 ± 15.25 | 368 | 581 ± 259.3 |
| | BR | 35.91 | | 57.20 57.42 | 56.37 | | 469 | |
| A4 | SS | 76.24 | | 55.62 53.13 52.77 | 53.84 | | 757 | |
| | GW | 58.89 | 61.0 ± 10.77 | 47.12 46.53 47.27 | 46.98 | 50.1 ± 3.85 | 470 | 538 ± 149.2 |
| | BR | 56.52 | | 54.12 53.03 53.15 | 53.43 | | 543 | |
| A5 | SS | 77.83 | | 11.85 12.27 11.51 | 52.07 | | 889 | |
| | GW | 56.26 | 59.0 ± 13.83 | 71.42 70.83 71.49 | 71.25 | 63.4 ± 10.60 | 278 | 428 ± 316.2 |
| | BR | 52.05 | | 23.35 24.12 23.32 | 53.82 | | 442 | |
| A6 | SS | 77.83 | | 11.85 12.27 11.51 | 52.07 | | 889 | |
| | GW | 56.26 | 58.7 ± 13.83 | 71.42 70.83 71.49 | 71.25 | 63.4 ± 10.60 | 278 | 423 ± 316.2 |
| | BR | 52.05 | | 23.35 24.12 23.32 | 53.82 | | 442 | |

*2.2. Biowaste Composting—Industrial-Scale Study*

Six piles were tested in the full technological cycle on the industrial scale (Figure 1). The process began by mixing and forming the piles on an outdoor square (A1, A2, and A4) or inside a roofed hall (pile A3) using a front loader. The A5 and A6 piles were outdoors for the first three weeks, after which they were moved into the composting hall. The dimensions of individual piles are listed in Table 3.

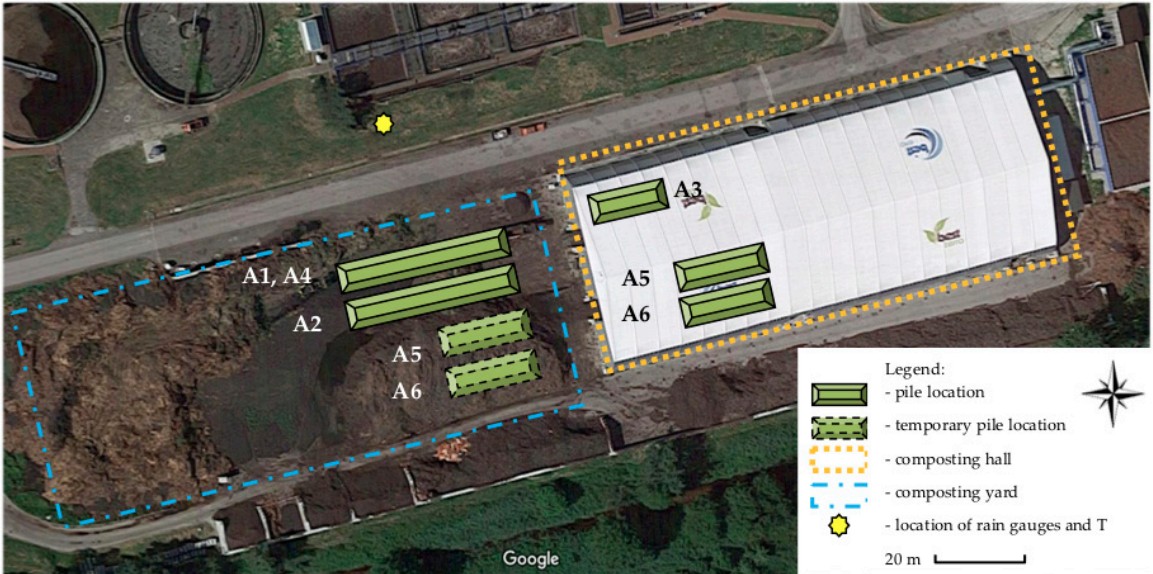

**Figure 1.** Composting plant layout. Wastewater treatment plant supplying sewage sludge for composting is located immediately to the north.

**Table 3.** The dimensions of tested compost piles.

| Pile | Height | Width | Length | Volume | Surface | Surface to Volume Ratio | H1 | H2 | H3 |
|------|--------|-------|--------|--------|---------|------------------------|----|----|----|
| | m | m | m | m$^3$ | m$^2$ | - | m | m | m |
| A1 | 1.5 | 5.0 | 23.5 | 152.1 | 307.1 | 2.0 | 0.750 | 0.75 | 2.0 |
| A2 | 1.8 | 3.0 | 33.0 | 144.3 | 306.2 | 2.1 | 0.625 | 0.90 | 1.5 |
| A3 | 2.1 | 5.5 | 17.0 | 168.4 | 264.4 | 1.6 | 0.750 | 1.05 | 2.0 |
| A4 | 1.5 | 4.2 | 22.0 | 116.8 | 250.6 | 2.1 | 0.625 | 0.75 | 1.8 |
| A5 | 1.6 | 3.5 | 15.5 | 69.9 | 158.1 | 2.3 | 0.625 | 0.75 | 1.5 |
| A6 | 1.5 | 3.7 | 16.0 | 72.3 | 167.1 | 2.3 | 0.625 | 0.75 | 1.5 |
| Average ± SD | 1.7 ± 0.24 | 4.2 ± 0.95 | 21.2 ± 6.67 | 120.6 ± 41.85 | 242.3 ± 65.70 | 2.1 ± 0.26 | 0.670 ± 0.060 | 0.83 ± 0.13 | 1.7 ± 0.25 |

Note: SD = standard deviation.

Pile aeration was passive. Turning was used for the A1, A2, A5, and A6 piles once or twice per week. Piles A3 and A4 were not turned (Table 4).

**Table 4.** The configuration of the tested compost piles.

| Pile | The Beginning of the Process | The Closing of the Process | Process Time (Days) | Turning Regime | Piles Location |
|------|------------------------------|----------------------------|---------------------|----------------|----------------|
| A1 | 04 July 2017 | 22 August 2017 | 50 | 2 times a week | Outdoor |
| A2 | 25 July 2017 | 12 September 2017 | 57 | 1 time a week | Outdoor |
| A3 | 22 August 2017 | 12 October 2017 | 52 | None | Indoor |
| A4 | 20 September 2017 | 09 November 2017 | 52 | None | Outdoor |
| A5 | 19 October 2017 | 07 December 2017 | 50 | 1 time a week | Outdoor/indoor |
| A6 | 19 October 2017 | 07 December 2017 | 50 | 1 time a week | Outdoor/indoor |

*2.3. Biowaste Sample Collection*

Samples of each biowaste type were collected from piles on the first and the last day during each weekly measurement. The second type of samples was taken using an excavator with three locations on the length of the pile. In each of them, the material was taken to represent the entire cross section of the pile (~10 samples weighing ~5 kg each), then it was combined into a cone and a final sample weighing ~15 kg was extracted using the quartering method [42]. The samples were transported in sealed plastic containers (without headspace) to the laboratory on the same day. The sum of precipitation from the

previous week was recorded from two rain gauges placed on the composting yard. The ambient and the composting hall temperatures were measured by the stationary thermometers (Browin 120308, Rybnik, Poland) and data were collected during the routine measurements of the chemical composition of gases in the piles (Figure 1).

### 2.4. Measurements of Process Gases and Temperature Distribution in the Piles

Gas samples were collected from 28 selected points during one test of each pile (i.e., four cross sections along the pile length, seven sampling points per cross section). The temperature was measured simultaneously at the same gas sampling points. The measurements of process gases and temperature distribution were conducted using a long probe made of stainless steel with holes at the end and a handle (Figure 2). A silicone tube was placed inside the probe through which gas was pumped from the probe to an electrochemical analyzer Kigaz 300 (Kimo Instruments, Chevry-Cossigny, France). The Kigaz 300 analyzer was factory calibrated by comparing it with standards of metrology laboratories.

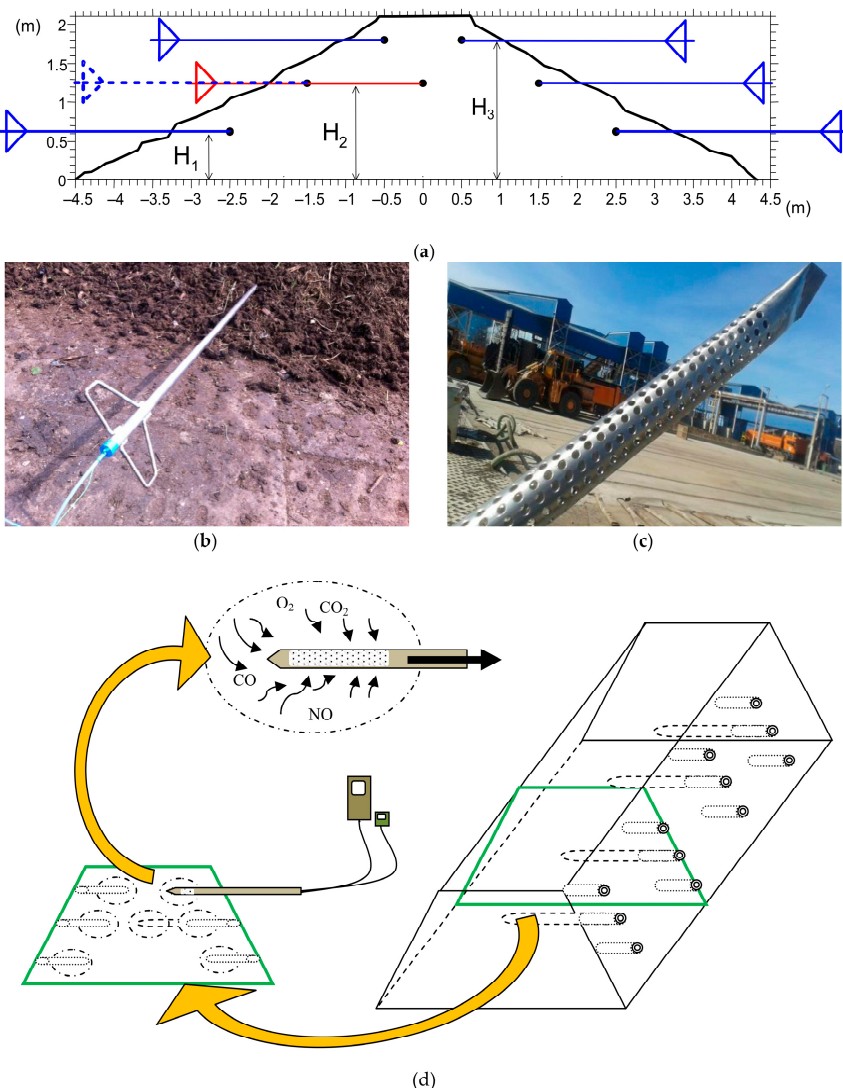

**Figure 2.** Cross section of a compost pile with the location of gas and temperature sampling points (**a**). The probe is introduced in pile (**a**); points 1–7 show the position of gas, and temperature sampling points ("shallow"—blue and "deep"—red). Probe for measuring the chemical composition of gases and temperatures in a pile (**b**). Perforated tip of the sampling probe to collect gases from the vicinity of the gas sampling points (**c**). Scheme of measuring gases in a pile (**d**).

Each time the analyzer was started, it performed about a 2-min autocalibration. After placing the probe in the measuring location, the gas concentration values were stabilized (typically within ~5 min). After each measurement, the probe was removed from the compost for ~1 min, allowing the measured gas values to return to ambient air (gas concentrations equal to 0%/ppm). At times it was necessary to restart the analyzer to perform auto-calibration again if ~1 min was not sufficient to return to ambient conditions. A Kigaz 300 analyzer is commonly used as a portable electrochemical device for the determination of composition exhaust (flue) gases from thermal processes. $O_2$ and $CO_2$ volumetric contents in piles were measured in % (±0.1%), but CO and NO contents were measured in volumetric ppm ±0.1 ppm.

A thermocouple measured the temperature of the probe tip. The temperature inside the piles was measured with ±0.1 °C precision. The entire measurement unit was made airtight to prevent atmospheric air from outside the pile getting into the probe (Figure 2d). The gas composition analyzer was equipped with an internal pump that created negative pressure, making it possible for the gas captured in the perforated zone to flow in and be directed to the analyzer.

### 2.5. The Spatial Arrangement of Measurement Points

Measurements were taken at four cross sections at 1/5, 2/5, 3/5, and 4/5 of the length of the pile. Seven measurements were taken at predetermined heights of the pile (H1 ~ 0.67 m, H2 ~ 0.83 m, H3 ~ 1.72 m) (Figure 2a and Table 3). The compost core measurements were at H2. All other measurements were approximately 1.50 m (horizontally) from the surface.

### 2.6. Data Analysis

Raw data were processed, summarized, and visualized for each cross section and each measurement time. Surfer 10 software (Golden Software, Cracow, Poland) was used to create color-coded isophlets to illustrate the spatial distribution of gas concentrations (CO, $CO_2$, $O_2$ and NO) and the temperature in piles. Four cross sections and two longitudinal cross sections (the left and right sides of the pile) were made for each of the six piles. In total 1375 cross sections were made and presented in 230 figures in a companion manuscript [43]. A small subset (16 cross sections) illustrates the chimney effect in the A1 pile and the temperature effect on CO distribution is presented in this manuscript.

## 3. Results and Discussion

### 3.1. Spatial and Temporal Distribution of Temperature within the Compost Piles

The analysis of the spatial and temporal distribution of temperature revealed several unexpected results. First, the occurrence of the chimney effect was not as prevalent as expected. Based on 183 temperature profiles collected in cross-sectional areas over the entire length of the study, only 8.2% of the temperature locations in a cross section developed the chimney effect. The temperature of composted biowaste was highest in the summer, especially on the south side of the piles located outdoors. In addition, only 31.1% of locations in a cross section reached thermophilic conditions (defined as T > 60 °C, necessary to kill pathogens). The entire dataset of the temporal and spatial variation of temperature is presented elsewhere [43].

There were a few exceptions (Figure 3), such as non-turned pile A3 (weeks 4, 6–8 only), where heat accumulation was observed in the central (core) part of the pile and the areas with higher temperature moved towards the apex (Figure 3). However, the temperature differences in the core and compost surface sections were smaller than those found in the literature, e.g., [44], wherein mesophilic conditions prevailed inside the compost after the 43rd day of the process, while the external zones were characterized by temperatures <10 °C. The reasons for the lack of a prevalent chimney effect in this research are many. There was no controlled and forced aeration, which could form a temperature gradient. Variations in the porosity of biowaste and environmental conditions could also affect the compost temperature.

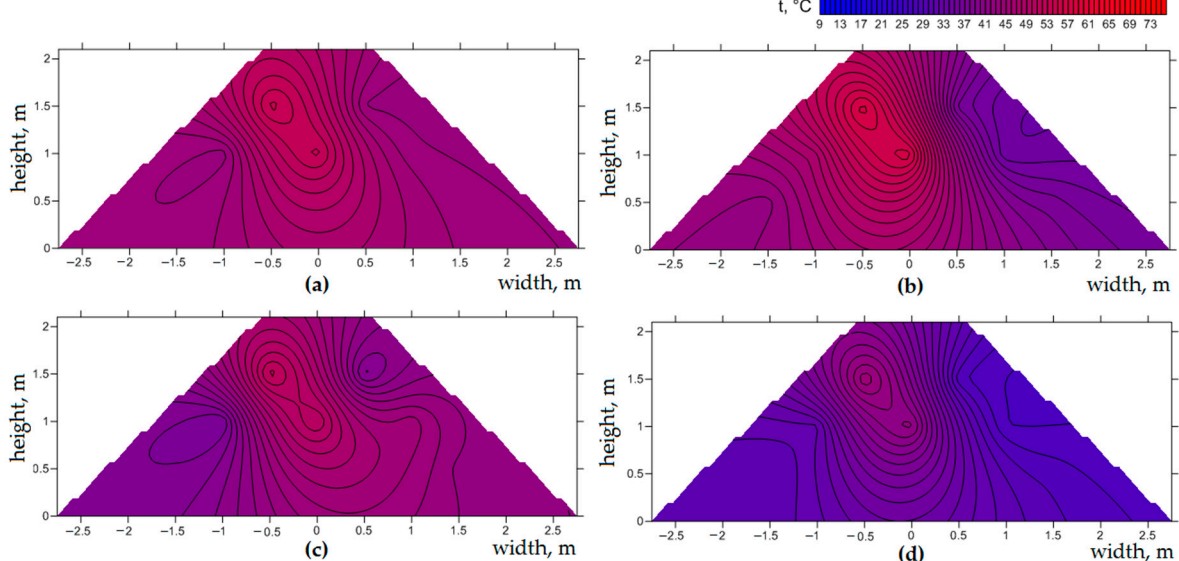

**Figure 3.** Chimney effect observed in pile A3: (**a**) on the 22nd day (week 4) of tests at the length of 11 m, (**b**) on the 36th day (week 4) at the length of 11 m, (**c**) on the 45th day (week 6) at the length of 16 m, (**d**) on the 52nd day (week 8) at the length of 16 m. Overall, the chimney effect was observed in 8.2% of the total cases studied.

Secondly, the apparent cooling of the compost material was different than that reported in the literature. In a wintertime experiment of composting with passive aeration [45], cooling began at the ends of the pile, which was explained by the increased surface area of the material and therefore increased heat loss. In this study, a similar pattern of temperature drop was observed only in piles A1 (36th day of the process, [43] (Figure A4)), A2 (15th and 23rd day, [43] (Figures A8 and A9)), and A3 (15th and 52nd day, [43] (Figures A17 and A22)), where it took place on just one end. Also, in some cases, areas that had previously cooled down reached a higher temperature at later stages of composting (e.g., the end of the left side of pile A6 at day 29 [43] (Figure A43)). It was also observed that low temperatures (prevailing in the piles in autumn and winter) were caused by an unfavorable surface-to-volume ratio, which resulted in faster heat loss (Table 3) [46]. In a similar study [31] of green waste composting, high temperatures (>60 °C) were observed even one year after the start of the process, with the piles larger compared to this study (i.e., 115 m long, 9 m wide, and 4 m high).

Thirdly, a cooler central part of the pile compared with the surrounding material was recorded in winter piles A5 (12th and 22nd day, [43] (Figures A33 and A34)) and A6 (2nd week of composting ([40] (Figure A40)). Fernandes et al. [47] explained the formation of this type of 'inversed' thermal gradient pattern by the appearance of areas that are rich in air necessary for oxidation initially in the outer parts of the pile and then progressing only later to the core. Fernandes et al. [47] also observed the occurrence of heat or low-temperature peripheries located at the base of the pile, the warmer core, and the zone of transition between the two. In this research cold peripheries also occurred, i.e., A2 pile during the whole process [43] (Figures A7–A14), A3 pile during the 1st-30th days and then on the 52nd day [43] (Figures A15–A19, A22); and in A4 pile at the 7th, 16th, 24th, and 30th days [43] (Figures A24–A27). The high level of heat loss from the bottom periphery of the pile resulted from exposure to the air, which surrounded it and caused the cooling of the material.

Fourth, the effectiveness of composting in winter varied, i.e., as far as consistently developing thermophilic conditions. Some piles (A5 and A6) reached thermophilic conditions despite low ambient temperatures (which in December fell below 0 °C, Table 5). In the case of pile A6 thermophilic conditions occurred within the first two weeks of the process in the whole pile profile ([43], Figures A39 and A40). In pile A5 in the same period, the temperature exceeded 60 °C only from the one side of the material in the central part of the longitudinal section ([43]; Figures A31 and A32). No apparent differences in A5

and A6 pile management can explain the observed difference in developing thermophilic conditions. Thus, caution needs to be exercised since this can have consequences for the elimination of pathogens in cooler piles such as A5.

**Table 5.** Ambient temperature and precipitation during composting process.

| Pile | Day | Date | Ambient Temperature | Precipitation | Average Temperature in a Pile |
|------|-----|------|---------------------|---------------|-------------------------------|
| | | | (°C) | (mm) | (°C) |
| | 1 | 04.07.2017 | 20.0 | no data | 31.6 |
| | 8 | 11.07.2017 | 21.0 | no data | 56.8 |
| | 15 | 18.07.2017 | 20.0 | 14 | 62.2 |
| | 22 | 25.07.2017 | 20.5 | 10 | 60.4 |
| A1 | 29 | 01.08.2017 | 36.0 | 6 | 59.1 |
| | 36 | 08.08.2017 | 25.0 | 2 | 61.9 |
| | 44 | 16.08.2017 | 28.0 | 10 | 46.3 |
| | 50 | 22.08.2017 | 20.0 | 18 | 47.6 |
| Average pile and ambient temperatures (±standard deviation)/precipitation | | | 23.8 ± 5.7 | 60 | 53.2 ± 10.7 |
| | 1 | 18.07.2017 | 20.0 | 14 | no data |
| | 8 | 25.07.2017 | 20.5 | 10 | 64.2 |
| | 15 | 01.08.2017 | 36.0 | 6 | 62.3 |
| | 23 | 08.08.2017 | 25.0 | 2 | 67.2 |
| A2 | 29 | 16.08.2017 | 28.0 | 10 | 51.1 |
| | 36 | 22.08.2017 | 20.0 | 18 | 52.7 |
| | 43 | 29.08.2017 | 22.0 | 0 | 47.1 |
| | 50 | 05.09.2017 | 14.0 | 27 | 40.6 |
| | 57 | 12.09.2017 | 14.5 | 22 | 28.4 |
| Average pile and ambient temperatures (±standard deviation)/precipitation | | | 22.2 ± 6.8 | 109 | 51.7 ± 13.1 |
| | 1 | 22.08.2017 | 20.0 | 18 | 37.0 |
| | 8 | 29.08.2017 | 22.0 | 0 | 57.8 |
| | 15 | 05.09.2017 | 14.0 | 27 | 53.9 |
| A3 | 22 | 12.09.2017 | 14.0 | 22 | 46.3 |
| | 29 | 19.09.2017 | 10.5 | 104 | 46.3 |
| | 36 | 26.09.2017 | 14.0 | 50 | 45.0 |
| | 45 | 05.10.2017 | 12.0 | 25 | 45.1 |
| | 52 | 12.10.2017 | 14.5 | 31 | 36.9 |
| Average pile and ambient temperatures (±standard deviation)/precipitation | | | 15.1 ± 3.9 | 277 | 46.0 ± 7.3 |
| | 1 | 20.09.2017 | 10.5 | 104 | 45.0 |
| | 7 | 26.09.2017 | 14.0 | 50 | 57.8 |
| | 16 | 05.10.2017 | 12.0 | 25 | 48.4 |
| | 23 | 12.10.2017 | 14.0 | 31 | 42.1 |
| A4 | 30 | 19.10.2017 | 9.5 | 0 | 36.9 |
| | 37 | 26.10.2017 | 13.5 | 16 | 33.9 |
| | 42 | 30.10.2017 | 4.5 | 35 | 26.0 |
| | 51 | 09.11.2017 | 4.5 | 15 | 22.2 |
| Average pile and ambient temperatures (±standard deviation)/precipitation | | | 10.3 ± 3.9 | 276 | 39.0 ± 11.8 |

**Table 5.** *Cont.*

| Pile | Day | Date | Ambient Temperature | Precipitation | Average Temperature in a Pile |
|------|-----|------|---------------------|---------------|-------------------------------|
|  |  |  | (°C) | (mm) | (°C) |
| A5 | 1 | 19.10.2017 | 9.5 | 0 | 52.8 |
|  | 8 | 26.10.2017 | 13.5 | 16 | 52.6 |
|  | 12 | 30.10.2017 | 4.5 | 35 | 32.1 |
|  | 22 | 09.11.2017 | 4.5 | 15 | 32.1 |
|  | 29 | 16.11.2017 | −1.5 | 21.5 | 35.8 |
|  | 36 | 23.11.2017 | 12.0 | 15 | 22.3 |
|  | 43 | 30.11.2017 | −1.5 | 22 | 16.6 |
|  | 50 | 07.12.2017 | 2.0 | 31 | 15.4 |
| Average pile and ambient temperatures (±standard deviation)/precipitation | | | 5.4 ± 5.8 | 156 | 32.5 ± 14.5 |
| A6 | 1 | 19.10.2017 | 9.5 | 0 | 60.0 |
|  | 8 | 26.10.2017 | 13.5 | 16 | 60.2 |
|  | 12 | 30.10.2017 | 4.5 | 35 | 27.6 |
|  | 22 | 09.11.2017 | 4.5 | 15 | 27.5 |
|  | 29 | 16.11.2017 | −1.5 | 21.5 | 19.3 |
|  | 36 | 23.11.2017 | 12.0 | 15 | 24.5 |
|  | 43 | 30.11.2017 | −1.5 | 22 | 19.3 |
|  | 50 | 07.12.2017 | 2.0 | 31 | 14.8 |
| Average pile and ambient temperatures (±standard deviation)/precipitation | | | 5.4 ± 5.8 | 156 | 31.7 ± 18.1 |

The highest temperatures and their fluctuations were observed in the A1 and A2 piles—nearly 70 °C in the 3rd week of composting. Both piles were placed on a composting yard during summer (Figure 1); however, they differed in terms of the frequency of turning (Table 4). In the case of high temperatures observed inside the piles, most were seen as a result of compost turning, and the composting process was carried out in the hottest months of the year—with the air temperature reaching 36 °C, combined with low precipitation (Table 5). The turning influenced the initial decrease and then the increase of temperature before the next pile turning, suggesting that turning stimulated the activity of microorganisms for the decomposition of organic substances. A similar trend was also observed by Beck-Friis et al. [48]. Joshua et al. [49] showed that the highest temperatures during turned green waste composting on an industrial scale (~70 °C) occur between the 3rd and 9th days of the process, which is consistent with the observations in this research.

An increase in the temperature of the compost pile during the early stages of the process was observed in all the piles. This was likely a result of the biochemical heat production and metabolism of microorganisms. This heat accelerates, in turn, the rate of metabolism of subsequent microorganisms, and thus the biodegradation of substrates present in the material and the production of heat [50]. High temperatures were observed during the 1st or 2nd week of the process in each of the analyzed piles. Temperatures gradually decreased after the initial two-week period (Table 2), likely due to the reduction of easily degradable substrates in piles. The same trend was observed previously by Wang et al. [50], investigating the spatial diversity of biological activity during the composting of sewage sludge. To a lesser extent, the peaking and then decreasing of temperatures was visible in the first three piles constructed in summer (A1–A3), where there were still areas with elevated temperatures during the final phase of the process. A decrease in post-peak temperatures in piles built in the autumn-winter (A4–A6), was observed in all cross sections at ~5–6 weeks, when the temperature dropped to 20-36 °C.

A clear trend of the influence of the ambient temperature on the temperature observed in piles was noted—the lower the ambient temperature, the lower the average temperature in a pile (Table 5). Also, the place of composting had an effect on the temperature inside the pile, particularly noticed in

the case of the A3 and A4 piles, which were composted at a similar time. Low ambient temperature (average 10 °C) and very high precipitation (276 mm) influenced the observation of lower temperatures in A4 (average 39 °C) and in A3 (average 46 °C) composted indoors (Table 5). On the other hand, in A3 a relatively constant temperature level was noticed, which, apart from the lack of turning, can also be explained by the location of the pile in the hall, where it was not exposed to sunlight.

After the initial two weeks of composting, a rapid decrease in the temperature inside the pile, to ~30 °C, was observed in A5 and A6. This trend can be explained by a sudden drop in ambient temperature after the 9th day of the process—from 13.5 °C to 4.5 °C (Table 5).

The applied method of aeration and atmospheric conditions also significantly influenced the weight reduction, which was the largest in the A1 pile (>60%). Other piles that were not turned or were composted in less favorable weather conditions (low ambient temperature, high precipitation; Table 5) achieved a lower average temperature in the process and showed a significantly lower weight reduction (Table 1). Observed high precipitation during the composting of the A5 and A6 piles caused a high relative humidity of air in the hall and contributed to the very low effectiveness of weight reduction or even weight increase (Table 1).

Due to the fact that the $O_2$ conditions and the temperature of the composted biowaste largely depended on the season, it is recommended to conduct a different technological regime for winter and summer. In winter, turning once a week for piles placed indoors is recommended. No turning is recommended for the winter and piles outside due to low outside temperatures that cause excessive cooling and moisture buildup. In summer, the most favorable recommended procedure is turning twice a week for the first two weeks, followed by weekly turning for the next two weeks. After that, turning can be stopped unless additional removal of moisture is needed. In this case, the turning should continue until the process is completed. The size of the pile should follow the surface-to-volume ratio: <2.5 in summer and <2 for cooler ambient conditions.

### 3.2. Spatial and Temporal Distribution of Oxygen and Carbon Dioxide

Compost piles were characterized by the occurrence of the largest number of areas with low $O_2$ concentration in the initial phase of the process. Low $O_2$ concentrations corresponded to the temperature approaching thermophilic conditions (typically the first 1-3 weeks). This is related to the increased activity of microorganisms that consumed oxygen in the decomposition of easily degradable substrates during composting of food waste in laboratory conditions [51]. The only exceptions were piles A1, where higher temperatures lasted until the 36th day of composting, and A3, which was characterized by a high degree of oxidation of the material throughout the process.

There was also an apparent correlation between the $O_2$ and $CO_2$ concentrations. Lower $O_2$ levels (due to its utilization and the decomposition of organic matter) were simultaneous with elevated $CO_2$ concentrations in each of the six piles (Figure 4) [21,52]. An anaerobic core and oxidized outer layer were visible in each of the six piles, especially in the initial phase. Some exceptions were noted, such as the 31.5 m cross section on the 56th day (Figure 5). This observation is consistent with the reported difference in $O_2$ concentration between the core and the top cover part of several percentage points in laboratory-scale composting on organic household waste [48]. Parr et al. [53] explained this by the lack of $O_2$-rich air to meet the aerobic needs of microorganisms, which Sommer and Moller [21] corroborate.

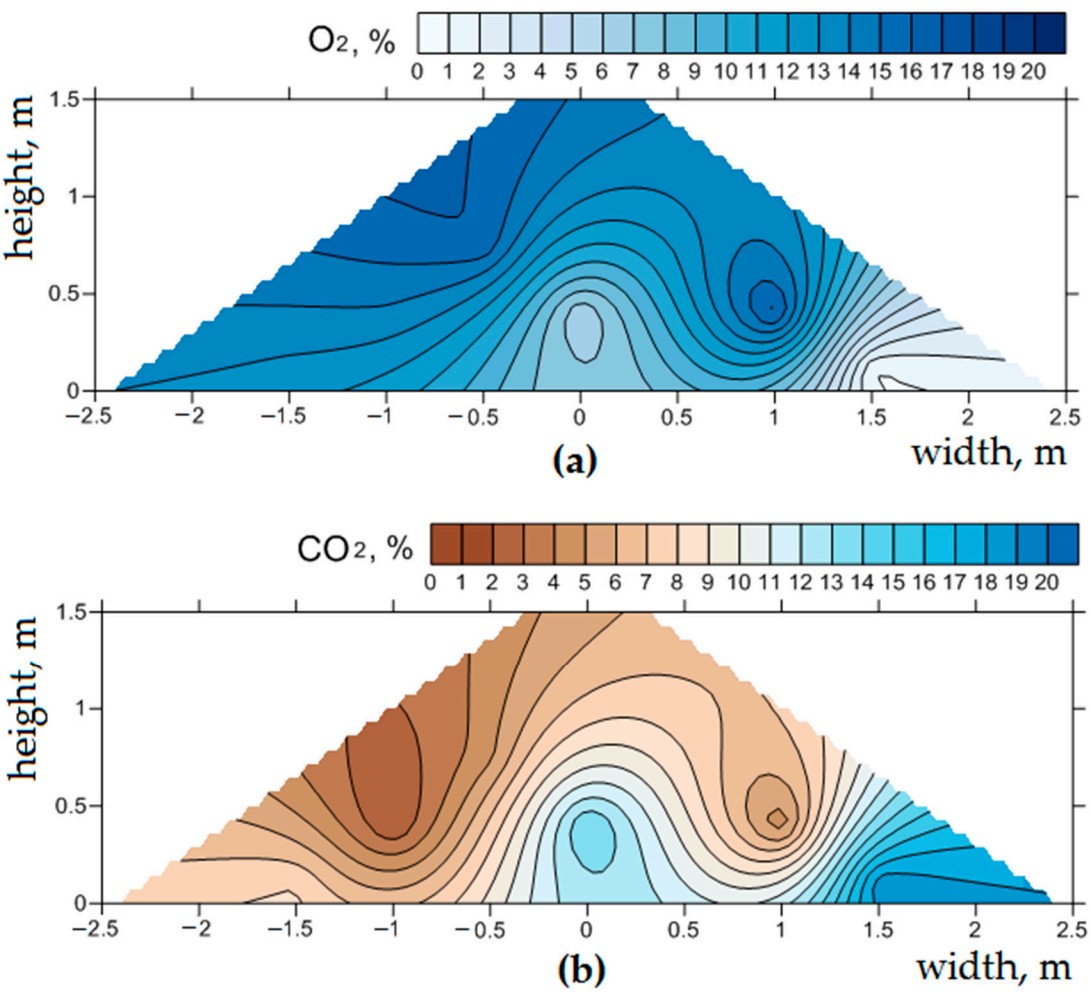

**Figure 4.** Spatial distribution of (**a**) $O_2$ and (**b**) $CO_2$ in the pile A1 on day 22 of tests at 22.5 m length.

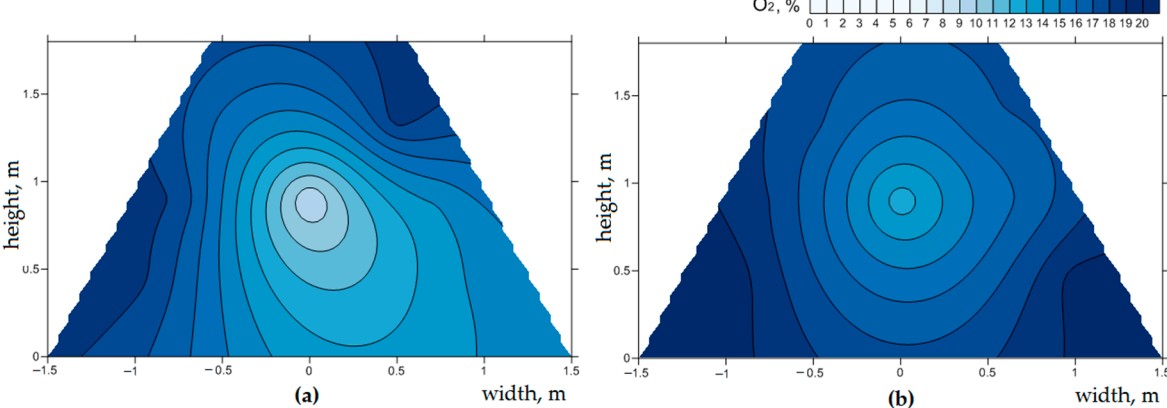

**Figure 5.** Spatial distribution of oxygen in pile A2: (**a**) on the 15th day of analysis at the length of 21.5 m, (**b**) on day 56 at the length of 31.5 m.

A surprising result was the high degree of oxidation of waste in the A3 pile throughout the process. Anaerobic conditions were expected due to the lack of material turning, heterogeneous conditions, and the fact that this pile had the largest width and height out of the six studied. According to Fukumoto et al. [54], the volume of anaerobic layers in the interior of the pile grew logarithmically in relation to the volume of oxygenated sites along with the increase of the pile (pilot-scale composting of swine manure). Despite these features, pile A3 reached a higher $O_2$ content than the smaller and narrower

pile A1, which was turned twice a week. Turning the material with such frequency should cause waste homogenization and interference with anaerobic conditions by breaking aggregates with low $O_2$ content. The expected results were observed to a greater extent in the second of the non-turned piles (A4), where, apart from the 39th and 42nd days of analysis, fragments with low $O_2$ concentration at a level of several percentage points were visible ([43]; Figures A69–A76).

In the early composting phase, $O_2$ was lost most rapidly from the top of the pile (e.g., days 15 and 23 in A2 [43]; Figures A54 and A55; 7th day in A4 [43]; Figure A70; and 8th day in A6, [43]; Figure A86). The $O_2$ loss started to incorporate the bottom of the cross sections (from the 22nd day in A5 and the 29th day in A6, [43]; Figures A80–A84, A89–A92) at later stages. Yamada and Kawase [55] observed the first of these trends during the composting of waste-activated sludge in a laboratory experiment during which the $O_2$ concentration decreased during the thermophilic phase with the increase of the distance from the bottom. The $O_2$ loss in the upper part of the piles was associated with the axial $O_2$ concentration gradient. The decreasing $O_2$ levels in the lower part of the pile could be caused by the displacement of gases from pores in the bottom material with leachates. Areas with increased $CO_2$ concentration were also recorded in the central (core) part of the piles, as well as at the base of the "arms" (far reaches) (e.g., 10.0 m of the length of pile A6 on day 1 of the process and its right side on day 22 [43]; Figures A85 and A88). This gas production was also reported as 'hot spots' present during the composting of cattle manure on a pilot scale [20], which in the case of $CO_2$ was explained by (1) its production in aerobic conditions and (2) the intensive anaerobic transformation of organic matter in these locations.

### 3.3. Spatial and Temporal Distribution of Carbon Monoxide

A characteristic feature of CO temporal distribution during composting in the A1–A6 piles was its increased production in the initial phase of the process. This agrees with the results of our earlier study [2] (i.e., composting of municipal solid waste on an industrial scale) and other authors [9] (i.e., decomposing organic waste and litter under laboratory- and pilot-scale composting plant conditions). The high CO production became apparent from the first week in the A3–A6 piles ([43]; Figures A153, A161, A169, and A177). The exception was the A1 pile, in which the increased CO concentration persisted until the 36th day ([43]; Figures A139–A142). Winter piles (A5 & A6) were characterized by the lowest CO content, which could be due to low ambient temperature. Haarstad et al. [3] made similar observations while composting organic waste in laboratory conditions, i.e., the lowest CO levels in piles prevailed in December, January, and March, when the outside temperature was <0 °C.

Increasing the concentration of CO in the initial period of the process correlated with the increasing temperature of composted biowaste. This occurred in five out of the six tested piles. The CO concentration increased under thermophilic conditions in pile A1 (day 8—Figure 6; 22nd and 36th day, [43]; Figures A141 and A142), A2 (8th, 15th, 23rd day [43]; Figures A145–A147), A6 (day 1 and 8, [43]; Figures A177 and A178), and lesser extent in pile A4 (day 7, [43]; Figure A162) and A5 (day 1, [43]; Figure A169). The non-turned A3 pile, composted indoors, had the highest CO levels recorded in colder conditions (1st day of the process, [43]; Figure A153), and when the temperature increased later, CO increase did not occur. This is in line with the results of laboratory-scale municipal solid waste compost reporting that the source of CO was thermochemical in nature [56].

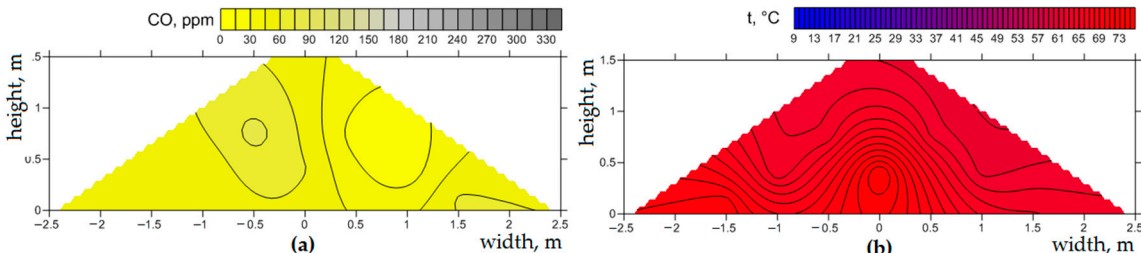

**Figure 6.** Spatial distribution of carbon monoxide in the pile A1 on the 8th day of analysis at the length of 1.5 m during the thermophilic phase (**a**) and temperature in the pile A1 on the 8th day of analysis at the length of 1.5 m (**b**).

The decrease in CO concentrations, which occurs in the later stages of composting, can be explained by two factors. The first is the consumption of CO by bacteria; their increased activity is likely to result in the oxidation of this compound to $CO_2$ [56]. The second is based on the dependence of the presence of CO on the $O_2$ concentration in the biowaste. The production of CO is the highest in aerobic conditions; the depletion of $O_2$ causes the CO level to decrease, which in turn influences the increase in the concentration of $CO_2$. This was evidenced by the spatial distribution of CO concentration in the compost pile, which was consistent with the $O_2$ gradient—the highest values were recorded near the top of the pile, while the lowest were in the lower part [9,31].

An inverse correlation between CO and $CO_2$ concentrations in compost piles was also observed, i.e., when one of the gases increases, the level of the second one falls [11]. However, the minimum CO concentration occurs when the maximum is reached by $CO_2$, which in turn can be evidence that the source of the CO is microorganisms [18]. The dual nature of CO production was also observed by Stegenta [10] (at both laboratory and technical scales), analyzing differences in the emission from sterilized and non-sterilized biowaste. The results confirmed that CO may be formed in a biotic or abiotic manner. In the first case, CO production is due to microorganisms active in the temperature range of 10-40 °C.

Unlike in the literature, no increase in CO concentration was observed after the biowaste was turned, likely due to the re-treatment of anaerobic fragments of material. A high concentration of CO sometimes occurred near the surface of the pile, but these areas always "overlapped" with lower $O_2$ content areas (i.e., the 1.5 m cross section in A2 pile on day 23 [43]; Figure A147). When the $O_2$ concentration increased, the CO was not detectable (e.g., the same pile, the entire profile, the 29th day of measurements, [43]; Figure A148). A similar trend in which CO occurs in anaerobic areas has been observed by Haarstad et al. [3]. This can be explained by the production of CO by methanogenic bacteria, which was observed during the measurement of gaseous emissions from different soils [12].

### 3.4. Spatial and Temporal Distribution of Nitric Oxide

NO concentration reached the highest level on the 1st day of the process in areas where the oxygen content was low in the A1, A5, and A6 piles (Figure 7). This can be caused by nitrogen loss in the form $NH_4$-N, as was also previously observed repeatedly during the composting of animal manure [57]. The location of increased NO content with low $O_2$ concentrations also indicates that denitrification processes are occurring. Nitrogen compounds are ubiquitous in almost every environment (both in mesophilic and thermophilic conditions), in the presence of $O_2$ and in anaerobic conditions. NO is a by-product of many processes, including nitrification, denitrification, and loss of N by leachate formation [57]. N losses in the initial phase of composting animal wastes are associated with process gas emissions and can reach 46.8-77.4% [58]. The same conclusion was reached during the composting of sewage sludge at a pilot scale by Witter and Lopez-Real [59], who observed the greatest decrease in the amount of this element in the gas form at the initial stage of the process.

For the A2–A4 piles, the NO concentration increased in the final phase of the process (43rd and 56th day in A2 [43], Figures A196 and A198; 36th and 52nd day in A3 [43], Figures A204 and A206; and

30th day in A4 [43], Figure A211). This could be due to several factors. First is the immobilization of inorganic N that occurs in the initial phase of composting [60]. It is incorporated into microbial biomass as the populations grow, when there is a large number of organic compounds in the material that are easily degraded [57]. The second explanation for the lack of areas of NO in the initial phase of composting is too high a temperature prevailing in the biowaste. Nitrifying bacteria are sensitive to high temperatures and do not appear in thermophilic conditions [61]. The nitrification in the initial stage of composting is inhibited not only by too high a temperature level but also by the production of $CO_2$ [62].

Many authors point out that the NO production changes when the temperature in a pile decreases to ~40 °C. Sanchez-Monedero [63], composting organic waste in a pilot plant, explained that the nitrification process depends not only on this temperature level but also on favorable aerobic conditions. This is consistent with our observations in the case of the A2–A4 piles, i.e., a higher NO concentration was associated with temperatures ~40 °C. In addition, Martins and Dewes [58] observed a higher release of NO from the top of the material, especially at the piles' "arms". It was also visible in A2 (cross section at 1.5 m on day 43 [43]; Figure A196) and A3 (right side 16.0 m in length on day 36 of the process [43]; Figure A204). This observation is consistent with those reported in [62] for the composting of animal waste. In the A4 pile, on the 30th day of analysis, NO accumulated in the lower part of the pile, which is visible at the length of 7.25 m ([43]; Figure A211). Unlike the previous two cases, the location of this area can be explained by the accumulation of nitrogen-containing leachate there.

Related studies composting pig slaughterhouse waste at a laboratory scale show the impact of turning on the dynamics of N transformation, which causes an inflow of $O_2$ and controls the processes of biodegradation, nitrification, and ammonification [64]. The material transfer causes N losses in the gas form, which is explained by the homogenization of the material and the improvement of its ventilation [58]. However, similar to the case of spatial $CO_2$ and $O_2$ distribution, it was not observed in piles A1–A6. In the A3 non-turned pile, NO occurred in negligible quantities from day 1 of the process until week 6. Similarly, in pile A4, NO was present throughout the composting period, although at a low concentration.

The spatial and temporal distribution of temperature, $O_2$, $CO_2$, CO, and NO could have been confounded by several factors such as the scale, biowaste type, environmental factors, and management. Therefore, a larger number of field tests could be carried out and the results used to extrapolate data to national estimates of emissions from the composting process. It should be mentioned that this research focused on the 'within' process. The gaseous emissions from the process were not measured and estimated. Future work could focus on linking the process with process gas emissions to the atmosphere and occupational exposure to workers involved in compost turning and instantaneous release of potentially toxic gases. A continuation of this research could also focus on developing the most effective and environmentally friendly method of composting process management that uses multiple inputs. Additionally, the mechanisms of CO and NO generation deserve further research focused on conditions (aerobic, anaerobic, thermophilic, mesophilic), perhaps first at the laboratory scale. They would allow a more accurate understanding of the nitrogen transformation in the composted material, which in turn would help us find a way to minimize CO and NO emissions from the process.

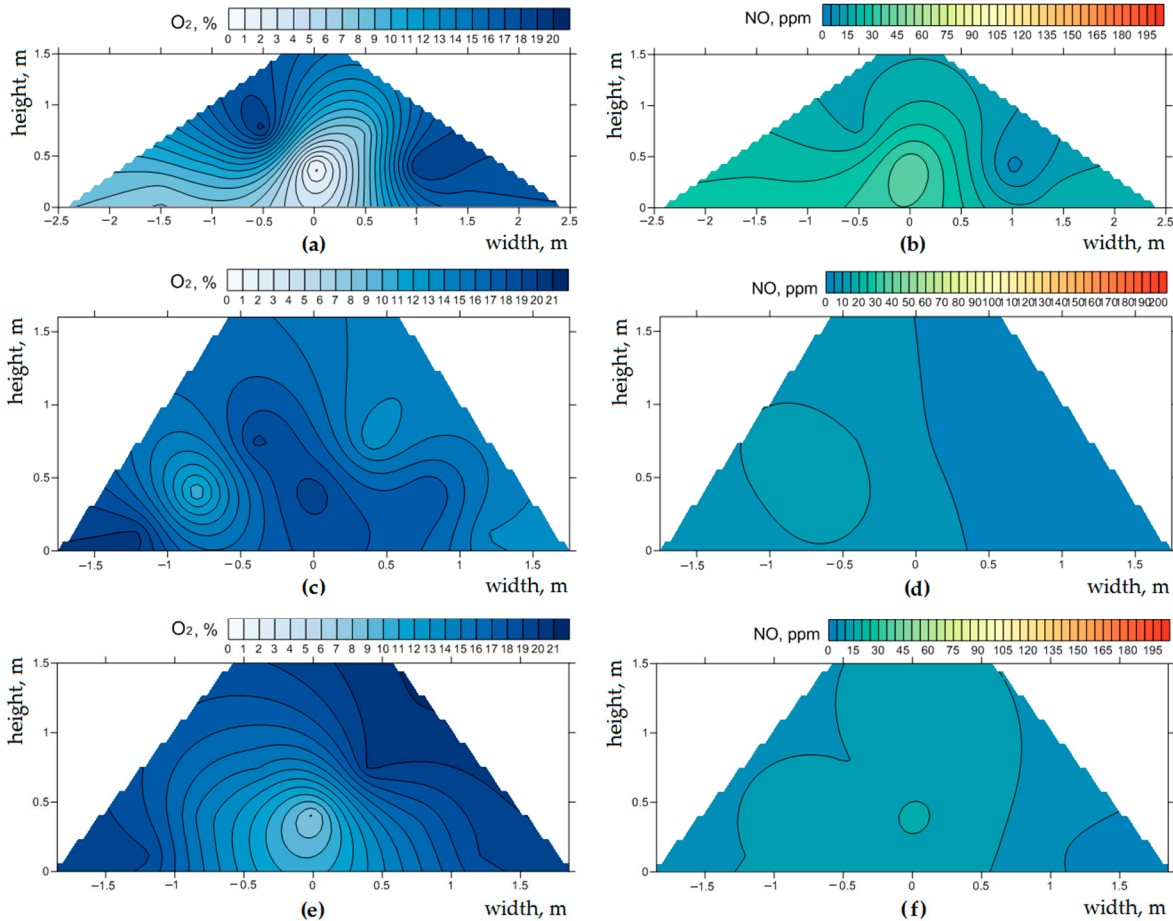

**Figure 7.** Spatial distribution of nitric oxide on the 1st day of analysis in comparison with the spatial distribution of oxygen: (**a**) oxygen concentration in pile A1 length 1.5 m, (**b**) nitric oxide concentration in pile A1 length 1.5 m, (**c**) oxygen concentration in pile A5 length 1.0 m, (**d**) nitric oxide concentration in pile A5 length 1.0 m, (**e**) oxygen concentration in pile A6 length 1.25 m, (**f**) nitric oxide concentration in pile A6 length 1.25 m.

### 3.5. The Efficiency of Biowaste Composting Processes

The initial organic matter (lost on ignition, LOI) in the compost was similar in all analyzed piles ~60% DM (Table S1, [43]). The final organic matter in the compost varied from ~35% DM in piles A1–A3 to ~45% DM in A4–A6 (Table S1, [43]). A higher efficiency of the process was therefore observed in piles that were turned and had a higher temperature inside the pile (Table 5). Turning of the pile stimulates microorganisms to degrade organic substances by changing the conditions to more favorable ones, i.e., the release of accumulated heat and toxic gases, and an increase in the $O_2$ saturation [65]. In each of the analyzed cases, a clear and continuous drop in the organic matter content in the piles was observed, whereas for piles A2–A4 the time needed to decrease the organics to below 40% DM was the shortest at ~40 days (Table S1, [43]).

Another useful index to evaluate waste stability is the respiration activity for four days ($AT_4$). Waste is considered stabilized when $AT_4$ drops below 10 mg $O_2 \cdot$g $DM^{-1}$ [66]. While the initial waste $AT_4$ value in piles A1-A4 was in the 41-67 mg $O_2 \cdot$g $DM^{-1}$ range, it was as low as ~30 mg $O_2 \cdot$g $DM^{-1}$ in piles A5 and A6 (Table S1, [43]). This is the result of seasonal variability in green waste, which in the summer is rich in easily biodegradable substances (mainly grass) but in autumn and winter is mostly composed of hardly decomposable substances (mainly leaves and branches). The time to reach the $AT_4$ threshold value <10 mg $O_2 \cdot$g $DM^{-1}$ also matters [66]. Thus, in A1 and A2, this time was shorter than 35 days, while in the other piles the stabilization time was extended to over 45 days.

Both the material and the weather conditions, including frequent and intense rain (in the case of waste composted outside), as well as air temperature, influenced the activity of microorganisms. On the other hand, high microorganism activity was also stimulated by the turning regime in the A1 and A2 piles. The course of biowaste stabilization had a similar character to municipal waste composting [67].

Different processing efficiency was also noted in the case of moisture removal, which influenced the effectiveness of total mass decrease. The A1 and A2 piles were characterized by a very high water removal rate (Table 1), which was due to the high ambient temperature (Table 5) combined with the high frequency of turning. However, pile A3 required a longer duration of composting to achieve moisture decrease, but in the case of piles A4–A6 an increase in the moisture in relation to the initial value was observed (Table S1, [43]). Such a situation resulted from the low efficiency of the process, caused by low ambient temperature and high air relative humidity close to 100% inside the hall (measurements from the factory reader placed inside the hall).

The Leopold matrix was used for the selection of technological recommendations [68] (Table 6). In the prepared matrix, the specific numerical values were assigned to each of the five parameters ($O_2$ concentration in a pile, temperature in a pile, final waste $AT_4$ index, final LOI, and moisture content in waste), defining the effectiveness of the composting process. The sum of the values of these five parameters was the basis for an indication of which variant of the process was the most effective. The results of the analysis were made on the basis of the data presented in Table S1 [43]. It was shown that A1 and A2 piles were the most effective, receiving 8 and 6 points, respectively, out of a maximum 10. The A3 and A4 piles were characterized by a much lower efficiency (0 points), and the most unfavorable conditions were in the A5 and A6 piles (−5 and −6 points, respectively) (Table 7).

**Table 6.** Assessment criteria of a matrix for evaluation of biowaste composting (Leopold method [68]).

| Point Scale | Time of Occurrence of the Temperature > 50 °C in a Pile | Time of Occurrence of $O_2$ > 10% in a Pile | Time to Reach $AT_4$ < 10 mg $O_2 \cdot g\ DM^{-1}$ | Time to Reach LOI < 40 % DM | Time of Occurrence of the Moisture < 40% |
|---|---|---|---|---|---|
| Unit | Week | week | Days | days | days |
| −2 | <1 | <1 | >45 | >55 | >50 |
| −1 | 2–3 | 2–3 | 40–45 | 50–55 | 45–50 |
| 0 | 3–4 | 3–4 | 35–40 | 45–50 | 40–45 |
| 1 | 4–5 | 4–5 | 30–35 | 40–45 | 35–40 |
| 2 | >5 | >5 | <30 | <40 | >35 |

**Table 7.** Summary scores of a matrix for evaluation of biowaste composting (Leopold method [68]).

| Pile | Time of Occurrence of the Temperature > 50 °C in a Pile | Time of Occurrence of $O_2$ > 10% in a Pile | Time to Reach $AT_4$ < 10 mg $O_2 \cdot g\ DM^{-1}$ | Time to Reach LOI < 40 % DM | Time of Occurrence of the Moisture < 40% | Sum |
|---|---|---|---|---|---|---|
| A1 | 1 | 2 | 2 | −1 | 2 | 6 |
| A2 | 1 | 2 | 2 | 1 | 2 | 8 |
| A3 | −1 | 2 | 0 | 1 | −2 | 0 |
| A4 | −1 | 2 | 0 | 1 | −2 | 0 |
| A5 | −1 | 2 | −2 | −2 | −2 | −5 |
| A6 | −1 | 2 | 0 | −2 | −2 | −3 |

## 4. Conclusions

This research provides a unique analysis of process gases and temperature evolution within a large composting plant for mixed green waste and municipal sewage sludge. Compost temperatures and process gas concentrations showed distinct spatial and temporal patterns. Final weight varied from 66% reduction to 7% gain. Only 8.2% of locations developed the desired chimney effect (utilizing natural buoyancy to facilitate passive aeration). Only 31.1% of locations reached thermophilic conditions (necessary to inactivate pathogens). Lower $O_2$ levels were simultaneous with the apparent production and elevated $CO_2$ concentrations. CO production increased in the initial composting phase. Winter

piles were characterized by the lowest CO content. The most varied was the NO distribution in all conditions.

The $O_2$ concentration was lowest in the central part of the pile and aeration conditions were good regardless of the technological regime used. Turning once a week was sufficient overall. Based on the results, the most favorable recommended procedure is turning twice a week for the first two weeks, followed by weekly turning for the next two weeks. After that, turning can be stopped unless additional removal of moisture is needed. In this case, the weekly turning should continue until the process is completed. The size of the pile should follow the surface-to-volume ratio: <2.5 and <2 for cooler ambient conditions.

**Supplementary Materials:** The following are available online at http://www.mdpi.com/2071-1050/11/8/2340/s1, Table S1. contains raw data of loss on ignition, moisture and $AT_4$ in waste samples during the composting process.

**Author Contributions:** Conceptualization, A.B., S.S. and G.P.; methodology, A.B., S.S. and G.P.; formal analysis, A.B. and J.A.K.; validation, A.B., G.P. and J.A.K.; investigation, S.S. and K.S.; resources, K.S. and S.S.; data curation, A.B. and K.S.; writing—original draft preparation, K.S. and S.S.; writing—review and editing, A.B., G.P. and J.A.K.; visualization, K.S.; supervision, A.B. and J.A.K.

**Funding:** This work was supported by the Best-Eko Sp. z o.o. (Poland) as the research program 'Selection of substrates based on BEST-TERRA compost and composting technology at the composting plant at Boguszowice sewage treatment plant', No. B090/0010/17. The authors would like to thank the Fulbright Foundation for funding the project titled "Research on pollutants emission from Carbonized Refuse Derived Fuel into the environment," completed at Iowa State University. In addition, this paper preparation was partially supported by the Iowa Agriculture and Home Economics Experiment Station, Ames, Iowa. Project no. IOW05556 (Future Challenges in Animal Production Systems: Seeking Solutions through Focused Facilitation), sponsored by Hatch Act and State of Iowa funds. The publication is financed under the program of the Minister of Science and Higher Education; Strategy of Excellence—University of Research; in 2018–2019 project number 0019/SDU/2018/18 in the amount of PLN 700 000".

**Acknowledgments:** We would like to thank the Przemysław Bukowski and Marcin Dębowski for providing an early version of the cross-section of a compost pile schematic with the location of gas and temperature sampling points (Figure 2a), photo of perforated tip of the sampling probe (Figure 2c), both of which were previously submitted as a part of the technical report: Białowiec A., Bukowski P. "Raport z badań intensywności zachodzących tlenowych przemian biologicznych w pryzmach do biostabilizacji frakcji podsitowej [Report on the intensity of the aerobic biological processes occurring in the prisms for biostabilization of the municipal solid waste undersize fraction]", contracted by Municipal Cleaning Company in Warsaw, Poland, to which it was submitted on September 2015.

**Conflicts of Interest:** The authors declare no conflict of interest. The funders had no role in the design of the study; in the collection, analyses, or interpretation of data; in the writing of the manuscript; or in the decision to publish the results.

**Data:** Supplementary data (e.g., Table S1) to this article can be found in [43]: Stegenta S., et al. The spatial and temporal distribution of process gases within the biowaste compost. *Data* **2019**, *4(1)*, 37. doi: 10.3390/data4010037.

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
