# Peer review of "Analysis of the Spatial and Temporal Distribution of Process Gases within Municipal Biowaste Compost"

_sustainability, doi:10.3390/su11082340_

Round 1
Reviewer 1 Report
This manuscript addresses that the effects of spatial and temporal distribution of agricultural wastes on the production of gases (mainly CO, CO2, NO and O2). This study would be interesting for readers of Energies journal, and it is enough to be accepted to the journal. The current issues and experimental data are nicely summarized and organized in the manuscript. There are some minor errors (grammatical, typos, formatting, and spacing errors) that should be corrected. The reviewer enjoyed to read this manuscript.
Author Response
We submit the detail response to reviewer's comment in the attached file.

Reviewer 2 Report
Interesting project relating to the study of the monitoring of process gases (CO, CO2, NO, O2) spatial and temporal evolution during large scale composting of biowaste sources. However some issues arise from the analysis of the paper which are presented in more detail below.
13 keywords? Focus more effectively the framework of the work.
The term “Experimental design” for section 2.2 is erroneous since it implies that the experiments carried out were designed by the establishment of a specific set of parameters with predetermined values for the evaluation of a quantified influence on an also predetermined set of responses through statistical tools. In this case, every one of the six experiments exhibit particular conditions. It is more correct to refer an experimental planning of each test conditions.
Throughout the article (with the exception of the abstract) the chemical formulas (gases for example) are consistently presented with the numbers at the level of the text. Correct this problem putting the numbers in the formulas in subscript.
Line 139 – 3,270 m3.day-1 ; “3” and “-1” in superscript
Table 1, Table 2 and Table 5 – Introduce lines to clearly separate the data relating to each pile.
Table 2 – The data presented in this table is confusing. Column 3 refers to moisture content? It seems that “Moisture” (what is the basis for this percentage?), “Loss on Ignition” and “Bulk Weight” are not correctly centred. The unit “kg.m-3” passes to a second line.
Table 5 – Why repeating the information to identify the averages? Anyway the order is incorrect. “Average” should be enough.
A huge amount of data was collected in very different conditions. Therefore it is difficult to make the connection between the few results presented and the conclusions. For example how does the results support the conclusion that the best solution is “turning twice a week for the first two weeks, followed by weekly turning for the next two weeks” if this particular solution was not even tested for any of the experiments carried out?
The composting process was successful? There is not presented sufficient data to assess the quality of the compost produced in each of the six experiments and how the different conditions could have affected the overall process. It is interesting to notice that for piles A5 and A6 (which appear to be almost replicas in winter conditions), the only data presented referring to the conditions of the final pile (weight reduction in Table 1) seems to be very different.
How does the measuring of gases contents in the pile correlate with the expected gases emissions?
Author Response

(The authors gave the same response as above.)

Reviewer 3 Report
This paper studied the process gas evolution within a large scale municipal compost, which is a well-established and historical technology. To enhance the scientific merit of this paper, the reviewer would suggest to add one more section comparing the pros/cons of compost and other similar technologies (e.g., landfill, anaerobic digestion, gasification, pyrolysis, and hydrothermal process), so that the readers could more appreciate the advantage (or drawback?) of using compost over other methods.
The title of this paper is very similar to the reference [40] (https://www.mdpi.com/2306-5729/4/1/37). The reviewer would strongly suggest the author to modify the title so that these two articles can be distinguished. In addition, please explain what is the major difference between this study and the reference [40].
The reviewer also noticed some mistakes/flaws that need to be addressed:
Please reduce the numbers of keywords. If the authors insist to have so many key words, please justify the reasons.
In Table 2, the significant number is not consistent.
In Tables 4 and 5, the format of the date is different. Please keep it consistent.
Fig. 3 and Fig. 6 can surely be further improved. It's very hard to read them in a black/white version. Also, what does the x- and y-axis mean? Moreover, the reviewer wonder if there is any change in the z direction?
Author Response

(The authors gave the same response as above.)

Round 2
Reviewer 2 Report
In the revised manuscript, the authors address suitably all the recommendations. I appreciate the effort, and now, I invite the authors to check their work carefully before resubmission, particularly, in taking special consideration to the English proofreading that has been carried out. Nevertheless I consider that after this step the manuscript will be ready for acceptance.